# Nanobiotics and the One Health Approach: Boosting the Fight against Antimicrobial Resistance at the Nanoscale

**DOI:** 10.3390/biom13081182

**Published:** 2023-07-28

**Authors:** Riya Mukherjee, Jasmina Vidic, Elcio Leal, Antonio Charlys da Costa, Carlos Roberto Prudencio, V. Samuel Raj, Chung-Ming Chang, Ramendra Pati Pandey

**Affiliations:** 1Graduate Institute of Biomedical Sciences, Chang Gung University, No. 259, Wenhua 1st Road, Guishan Dist., Taoyuan City 33302, Taiwan; vhimanshu150497@gmail.com (H.); riya.mukherjee1896@gmail.com (R.M.); 2Master & Ph.D. Program in Biotechnology Industry, Chang Gung University, No. 259, Wenhua 1st Road, Guishan Dist., Taoyuan City 33302, Taiwan; 3Micalis Institute, Université Paris-Saclay, INRAE, AgroParisTech, 78350 Jouy-en-Josas, France; jasmina.vidic@inrae.fr; 4Laboratório de Diversidade Viral, Instituto de Ciências Biológicas, Universidade Federal do Pará, Belem 66075-000, PA, Brazil; 5Instituto de Medicina Tropical, Universidade de São Paulo, São Paulo 05403-000, SP, Brazil; 6Laboratório de Imunobiotecnologia, Centro de Imunologia, Instituto Adolfo Lutz, 351, São Paulo 01246-902, SP, Brazil; 7Centre for Drug Design Discovery and Development (C4D), Department of Biotechnology & Microbiology, SRM University, Sonepat 131 029, Haryana, India; 8Department of Medical Biotechnology and Laboratory Science, Chang Gung University, No. 259, Wenhua 1st Road, Guishan Dist., Taoyuan City 33302, Taiwan; 9Laboratory Animal Center, Chang Gung University, No. 259, Wenhua 1st Road, Guishan Dist., Taoyuan City 33302, Taiwan

**Keywords:** antibiotic, antimicrobial resistance (AMR), One Health, economic, nanotechnology, policy

## Abstract

Antimicrobial resistance (AMR) is a growing public health concern worldwide, and it poses a significant threat to human, animal, and environmental health. The overuse and misuse of antibiotics have contributed significantly and others factors including gene mutation, bacteria living in biofilms, and enzymatic degradation/hydrolyses help in the emergence and spread of AMR, which may lead to significant economic consequences such as reduced productivity and increased health care costs. Nanotechnology offers a promising platform for addressing this challenge. Nanoparticles have unique properties that make them highly effective in combating bacterial infections by inhibiting the growth and survival of multi-drug-resistant bacteria in three areas of health: human, animal, and environmental. To conduct an economic evaluation of surveillance in this context, it is crucial to obtain an understanding of the connections to be addressed by several nations by implementing national action policies based on the One Health strategy. This review provides an overview of the progress made thus far and presents potential future directions to optimize the impact of nanobiotics on AMR.

## 1. Introduction

Antibiotics have transformed modern medicine by revolutionizing the treatment of infectious diseases (IDs). Given that antibiotics are one of the foundations of modern medicine, there is currently no simple solution to the problem of pathogenic microorganisms acquiring resistance to antibiotic therapy due to indiscriminate use, overuse, and, in some cases, abuse of antibiotics over time [1]. Antibiotics used to treat bacterial infections can lose their effectiveness over time or lead to the development of antibiotic-resistant (AMR) organisms [2]. Furthermore, the modern convenience of mobility of products and infected persons aids in the spread of viruses at an unprecedented rate [3]. AMR infection, on the other hand, impacts the worldwide socioeconomic situation. Estimating the exact economic cost of resistant bacterial diseases remains a major global challenge. AMR is a substantial economical burden for the entire world. It will be more challenging for the poorer countries to deal with this circumstance. If no action is taken to tackle AMR infection, the socioeconomic situation around the world will suffer. The economy will fall, social inequality will rise, and the health care industry will become less sustainable. As a result, AMR reduction may reduce global GDP loss. Therefore, AMR is integrated into several of the goals in achieving the 2030 UN Sustainable Development Goals [4].

Being a new area of technological innovation, nanotechnology is essential in therapy, cost-effective prevention, and the development of diagnostic instruments. As a result of the rapid advancement of nanotechnology, nanomaterials have gained significant attention in biomedical applications. Nanomaterials have been identified as promising weapons [5] for many therapies due to their nano-size, high specific surface area, and abundance of modification sites [5,6]. By modifying the size, shape, and/or surface chemistry of nanoparticles (NPs), their functions can be adjusted to match specific requirements [7,8]. Engineered NPs offer significant promise as a viable alternative for treating various infections, particularly those caused by multi-drug-resistant (MDR) bacteria. The antibacterial properties of engineered NPs primarily involve their binding to the bacteria’s surface, ion release, and subsequent generation of high oxidative stress. This makes it challenging for bacterial cells to develop multiple simultaneous gene mutations to counteract NP-mediated treatments effectively. As a result, engineered NPs present a potential solution to address the problem of antibiotic resistance [9,10]. Numerous types of metal and metal oxide nanoparticles, including silver (Ag), silver oxide (Ag_2_O), gold (Au), titanium dioxide (TiO_2_), copper oxide (CuO), zinc oxide (ZnO), calcium oxide (CaO), silica (Si), and magnesium oxide (MgO), have been identified for their antimicrobial properties [11]. In addition, these NP used as antimicrobial coatings and wound dressings [9,12]. Currently, the medical sector has investigated NPs longevity, efficiency, durability, adaptability, and unique physicochemical properties. The most promising technique for dealing with AMR bacteria would be the combination of nanotechnology and antibiotics [13]. The combination of NPs loaded with conventional antibiotics results in a synergetic effect, where dual antibacterial action is exerted through the combined impact of both the nanoparticles and antibiotics [14]. They are being used in a variety of therapeutic approaches to combat AMR, including reduced toxicity and improved stability; targeted delivery to infection sites; stimuli-sensitive drug release; targeted towards biofilm microenvironments; and combined physical therapy photothermal therapy (PTT) and antibacterial photodynamic therapy (aPDT) [15,16].

Apart from that, bacterial isolates demonstrate synergistic co-resistance between heavy metals and antibiotics through similar mechanisms [17]. Research findings indicate that co-resistance of antibiotic resistance with heavy metals occurs through similar functional and structural mechanisms, which can be carried on plasmids or chromosomes. Antibiotic resistance genes (ARGs) naturally exist in various environments but are typically present in low amounts [18]. However, when certain pollutants such as heavy metals, crude oil, and sewage are introduced, the abundance of these ARGs increases. In such situations, these genes can be transferred between microorganisms through both horizontal and vertical mechanisms, leading to multi-drug resistance and exacerbating the clinical outcomes of infectious diseases [19]. Moreover, these environmental ecosystems can act as reservoirs for ARGs, facilitating their crossover from environmental settings into clinical environments [20].

One Health is an interdisciplinary field that connects humans, animals and the environment. The origins of One Health are centuries old and are founded on the mutual dependence of humans and animals and the awareness that they share not only the same habitat but also numerous infectious diseases [21]. Antimicrobial use involves various stakeholders, encompassing physicians, nurses, pharmacists, patients, their relatives, medical representatives, distributors, pharmaceutical companies, regulators, and policymakers. Additionally, stakeholders engaged in nonhuman use, such as those in the animal and agricultural industry, play a significant role [22]. Numerous prominent organizations, including the Centers for Disease Control and Prevention (CDC), the Infectious Diseases Society of America, the World Economic Forum, and the World Health Organization (WHO), have been working under this approach to combat AMR at the global level and proclaimed that AMR to be a “global public health concern” [23]. The WHO developed a global action plan (GAP) and Antimicrobial Stewardship Programs (ASP) to combat AMR. Implementing ASP comes with a set of challenges, including insufficient acceptance from physicians, inadequate knowledge about antibiotic prescribing, a shortage of staff for ASP activities, limitations in diagnostic testing to guide ASP interventions, and constraints in antibiotic choice due to the unavailability of certain antibiotics [24]. In this review, we focused on nanotechnology and the One Health approach to combat AMR for economic development. 

## 2. Antibiotics and AMR

The remarkable rise of AMR among pathogenic bacteria poses a significant threat to human health. Antibiotics are medications that have unquestionably helped humans tackle a wide range of microbial illnesses. Antibiotics have been widely used for therapeutic indications not only in humans but also in animal husbandry and agriculture domains for many years [25]. Antibiotic resistance poses significant public health concerns as the bacteria resistant to antibiotics found in animals can potentially be harmful to humans. These bacteria can easily be transmitted to humans through the food chain and can spread widely in the environment through animal wastes [26]. The interconnection between the environment and animals plays a crucial role in shaping public health outcomes. Notably, the soil and water environments are considered essential reservoirs and origins of antibiotic resistance [27], particularly as they are influenced by agricultural practices [28]. However, a study done by Lipsitch et al. [29], suggested the three possible ways to transmit the MDR gene from agriculture to humans. Firstly, a person contracts a resistant pathogen originating from agriculture either by coming into contact with livestock consuming bacteria-contaminated fodder or water. However, there is no ongoing transmission of the pathogen among humans in this scenario. Secondly, a person becomes infected or carries a resistant microbe through direct or indirect transfer, and this is followed by continued transmission among humans, with some individuals falling sick. This situation involves a “species barrier” breach by a microbe that might be directly harmful to humans or a commensal microbe with the potential to cause opportunistic infections. Lastly, resistance genes that emerge in agricultural settings are transferred horizontally into human pathogens. The resulting resistant lineages are then favored and selected due to the use of antibiotics in humans [29]. Many antibiotics were created in the twentieth century to treat bacterial infections (an overview has been illustrated in Figure 1). As a result, microorganisms adapted to growing amounts of antibiotics in the environment in response to AMR. Humanity is currently confronted with a rising number of AMR pathogens [30]. Antibiotics work on bacteria by interfering with crucial survival processes such as cell wall formation and suppressing the creation of important proteins, DNA, and RNA. Bacteria, on the other hand, have the inherent ability (developed from millennia of competition) to adapt rapidly through mutations and DNA transfer (through horizontal gene transfer) and it can also arise through diverse mechanisms [31], such as mutations in acquired genes, modifications leading to impermeable antimicrobial targets, enzymatic degradation or hydrolysis, and bacteria living in biofilms [32]. In addition bacterial biofilms and intracellular bacteria also contribute greatly to AMR due to drug delivery barriers, preventing the entry of drugs, expelling drugs through active efflux, mutating drug targets, and enzymatically inactivating drug function [33,34] to counter the threat posed by these antimicrobials. The inappropriate and excessive use of antibiotics greatly encourages such changes [35]. Several drug resistance genes from various organisms can be acquired by the same bacterium, creating a “superbug” that is multi-drug resistant (MDR).

The ID load in India is among the greatest in the world. A recent investigation revealed that antimicrobial drugs were being used inappropriately and irrationally against these infections, leading to a rise in the development of AMR. This situation prompts serious public health concerns, and an action plan to combat AMR is deemed important [36]. In Taiwan, the rise in antimicrobial-resistant Gram-negative bacteria has resulted in a significant increase in infections linked to poorer patient outcomes [37]. Regarding public health, it is essential to keep an eye on current situational analyses in the Indian and Taiwan settings so that suitable interventions can be launched at the community level to address the problem.

AMR has risen to the top of the list of public health concerns in the twenty-first century in both countries. Therefore, because of their unique physiochemical features, NPs can overcome AMR mechanisms, allowing nanomaterials to execute several novel bactericidal routes to achieve antimicrobial efficacy and provide a versatile platform to generate novel therapeutic strategies. Nanomaterials are close in size to bimolecular and bacterial cellular systems, allowing for more multivalent interactions than small molecule antibiotics [38,39]. A growing range of NP variations and NP-based products are being exploited as a new line of defense against microbial resistance and MDR. Different forms of NPs have different strategies for AMR. One commonly acknowledged connection between nanomaterials and their ability to fight bacteria is that they show promise as a supplement to antibiotics. This approach is becoming increasingly popular as it has the potential to address the limitations of antibiotics [9]. Furthermore, as an excellent transporter, nanomaterials can assist existing antibiotics [40].

## 3. Role of Nanobiotics in Overcoming the Challenge of Antimicrobial

A method of reducing antibiotic toxicity is to optimize their pharmacokinetics, notably by encapsulating the drug in NPs. Many of the alternative strategies such as addressing antimicrobial-resistant bacteria, targeting on antimicrobial-resistant enzymes, developing drug delivery systems, utilizing physiochemical methods and exploring unconventional strategies [41] proposed to tackle MDR bacteria use nanotechnology to create novel nanomaterials with broad-spectrum antibacterial activity [42]. NPs are promising since they contain bactericidal properties as well as the ability to deliver conventional antibiotics [43], and the different categories and their effectiveness are depicted in Table 1. Furthermore, several techniques are used for efficient NP delivery illustrated in Figure 2.

**Table 1 biomolecules-13-01182-t001:** Categories of nanomaterials and their effectiveness in combating bacteria.

Type of Nanomaterial	Nanoparticle	Particle Size (nm)	Shape	Mode of Action	Target Bacteria	Factors Affecting Antimicrobial Activity	References
**Organic**	Chitosan	200 nm	Spherical, distinct and regular shape	- Block exchange of nutrients, causing cell death. - Unprotonated amino groups chelate metal ions on the cell surface to disrupt cell walls.	- *E. coli* - *S. aureus* - *Shigella* - *Vibrio cholerae*	Concentration, pH	[44,45]
	Quaternary Phosphonium or Sulfonium Groups	1–100 nm	Polymorphic shape	- Inhibits bacterial growth. - Amphiphilic nature effect bacterial membrane lead to bacterial lysis.	- *E. coli* - *S. aureus*	Particle size Shape	[46,47]
	Poly-ε- lysine	1–100 nm	Star shape	- Disintegrate cell wall. - Disrupt bacterial membrane integrity.	- *E. coli* - *S. aureus* - *S. cerevisiae* - *L. monocytogenes*	Particle size Concentration	[48,49]
	Quaternary Ammonium compounds	1–100 nm	Spherical shape	- Disrupt the function of cell membrane. - Cell wall lysis due to autolytic enzymes.	- *E. coli* - *L. monocytogenes* - *Pseudomonas*	Particle size Concentration	[49,50]
**Inorganic**	Silver nanoparticle	5–100 nm	Sphere, disk or triangular shape	- Disrupt the bacterial cell membrane structure and inhibit the action of certain membranous enzymes	- *E. coli*	Morphology	[51,52,53,54]
	Gold nanoparticle	1–100 nm	Spherical, star and flower shaped	- Generation of cell wall apertures - Decrease ATPase activity - Bacterial membrane disruption	- Methicillin-resistant *Staphylococcus aureus*	Particle size	[55,56,57]
	Copper nanoparticle	2- 350 nm	Spherical-mono-dispersed shape	- Attach to the bacterial cell wall and permeate through cell membrane cause cell death	- *E. coli* - *Proteus vulgaris*	Particle size Concentration	[58,59]
	Zinc oxide NP	1–100 nm	Spherical, rods, needles, and platelets	- Cell surface adsorption - Damage of lipid and protein - Membrane disruption - Generate oxidative stress inside bacterial cell	- *E. coli* - *K. pneumonia* - *S. aureus* - *C. jejuni*	Particle size Concentration Morphology	[60,61,62]
	Magnesium oxide NP	15–100 nm	Spherical and crystalline structure	- Electrostatic interaction - Lipid peroxidation. - Alkaline effect	- *E. coli* - *Staphylococcus aureus* - *B. subtillus*	Particle size Concentration pH	[63,64]
	Titanium oxide NP	30–45 nm	Spherical	- Adsorption to the cell surface	- *E. coli* - *S. aureus* - *E. faecium*	Particle size Stability Concentration	[65,66]
	Iron Oxide	10–20 nm	Rod and spherical shaped	- Reactive oxygen species (ROS) induce oxidative stress, which damages proteins and DNA	- *B. subtilis* - *S. aureus*	Particle size	[67,68]
**Carbon Based**	Carbon nano tubes	1–100 nm	Spiral or rod shape	- Damage bacterial membrane - Inhibit energy metabolism - Damage respiratory chain - Physical interactions	- *E. coli* - *S. aureus* - *S. enetrica* - *Y. pestis*	Particle size Intrinsic properties	[11,69]
	Graphene nanoparticles	12 nm	Ellipsoid shape	- Disruption of bacteria cell membrane - Have strong entrapping ability - Oxidative stress caused due to ROS	- *Staphylococcus aureus* - *E. coli*	Particle size Particle shape	[70,71]
	Fullerenes	200 nm	Ball shape	- Outer membrane damage	- *E. coli* - *B. subtilis*	Particle size	[72,73]
**Composite**	Metal matrix	1–100 nm	Rod, peanut, star shaped	- Form irregular pore in bacteria outer membrane - Stop bacterial growth	- *E. coli* - *S. aureus* - *K. pneumonia*	Particle size	[74,75]
	Polymer matrix	5–100 nm	Spherical, rod shaped	- Stop bacterial growth and physical interaction	- *E. coli* - *S. aureus* - *K. pneumonia*	Depend on medium content	[76,77]
	Ceramic matrix	1–100 nm	Cylindrical shapes	- Stop bacterial growth - Highly antimicrobial	- *E. coli* - *S. aureus*	Particle size	[78,79]

**Figure 2 biomolecules-13-01182-f002:**
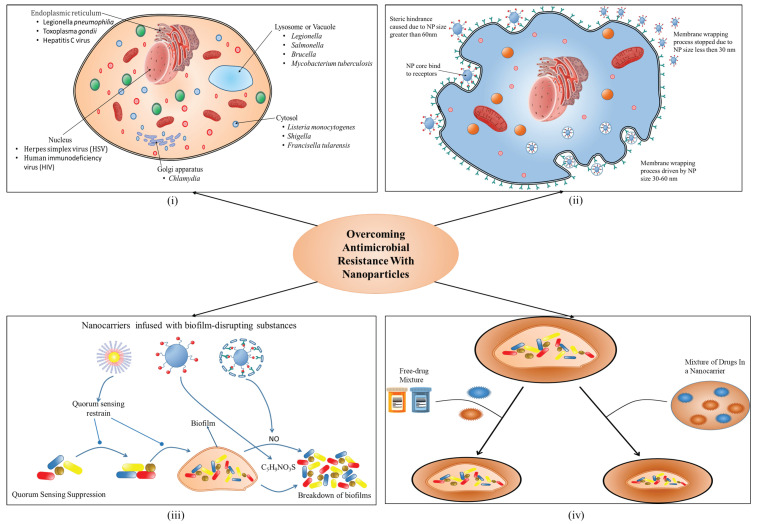
Using nanocarriers as a strategy to overcome resistance to drugs and illustrating different mechanisms of interaction. (**i**) Pathogens that live in intracellular foci that are difficult for medications to reach can avoid the effects of anti-infective [80]. (**ii**) Membrane-wrapping process affected by the different nano-particle size [81,82]. (**iii**) Moiety-containing nanocarriers that interfere with quorum sensing or the biofilm matrix can be placed onto them or surface functionalized. (NO = nitric oxide and C_5_H_9_NO_3_S = N-acetyl cysteine). (**iv**) Co-administration of the free-drug combination, co-encapsulation of medication combinations in nanocarriers as a potential approach to yield increased efficacy [83].

### 3.1. Reduced Toxicity and Enhanced Stability

Antibiotics will be protected by nanocarriers after being encased in polymeric NPs. As a result, the overall toxicity of antibiotics can be diminished. For example, Ori Baber and coworkers investigated the impact of airborne magnetic nanoparticles (MNPs) on BEAS-2B cells in vitro. Uncoated iron oxide was compared to two amorphous silica-coated MNPs, demonstrating that amorphous silica-coated MNPs resist acidic erosion [84]. As a result, particle stability results in less cytotoxicity and biological effect on human airway epithelial cells. Another example is polymyxin B (PMB); it has good bactericidal activity and is considered the last line of defense against Gram-negative bacteria. However, with systemic delivery, PMB causes severe nephrotoxicity and neurotoxicity. To address this issue, several PMB-loaded polymeric NPs were created in order to lower the systemic toxicity of PMB [85,86]. Vancomycin (VAN) and curcumin loaded NPs were synthesized for the same purpose [87,88]. Certain antibiotics can be degraded in vivo encapsulating antibiotics into NPs is an efficient method for improving antibiotic stability in vivo. This method can also be employed to administer antibacterial peptides (AMPs). Due to enzymatic breakdown, in vivo administration of AMPs is a significant issue. The NPs were subsequently treated with AMPs for improved in vivo stability [89]. 

### 3.2. Targeted Delivery to Sites of Infection 

Antimicrobial drugs can be targeted to the site of infection using NPs, allowing greater antibiotic doses to be administered at the infected site, overcoming existing resistance mechanisms with less harmful effects on the patient. Sometimes it is also regarded as on-demand drug delivery [90]. As with NPs targeting intracellular bacteria, NPs targeting the site of infection can release high concentrations of antimicrobial drugs at the site of infection while keeping the total dose of drug administered low, as illustrated in Figure 2 [91,92]. 

NPs can be targeted to sites of infection passively or actively. Improved on-target accumulation arises due to increased nanocarrier permeability at infection sites relative to uninfected tissue. Ligands (for, e.g., antibodies) that bind to diseased tissues or microorganisms as receptors (e.g., antigens) can also be functionalized on nanocarrier surfaces. This latter approach is known as active targeting or ligand-mediated targeting. Ligand-conjugated nanocarriers can also improve cell absorption, which can help treat intracellular infections. Rapid breakthroughs in targeted drug delivery systems, mostly in cancer *Mycobacterium tuberculosis* treatment, have occurred in recent years [7,92,93,94,95]. In the context of infectious disease, a rifampicin derivative has been linked to an anti-MRSA antibody in a mouse model to treat systemic MRSA infections [96]. In comparison to the free antibiotic, the conjugate was more efficient. Most crucially, the targeted therapy appeared to reach intracellular bacteria that would otherwise have remained ‘hidden’ from the antibiotic, becoming a latent cause of recurring illness. Passive targeting, on the other hand refers to strategies that do not rely on specific ligands [82]. Passively targeted NPs selectively extravagate at infection locations where inflammation has enhanced blood vessel permeability. Many elements, including hydrophobicity, van der Waals forces, and static electrostatic attraction, directly impact this form of delivery. The electrostatic interactions between the negative charge of the bacteria surface and the cationic charge of the NP’s surface can improve its efficacy [97,98].

### 3.3. Stimuli-Sensitive Drug Release

When antibiotic-loading NPs are utilized for infection control, antibiotics must be properly released from the NPs upon delivery at infected locations. Otherwise, antibiotics’ bactericidal efficacy will be significantly reduced. Several stimuli-sensitive drug nanocarriers were developed, which might provide selective drug release in infected tissues by taking advantage of the particular microenvironment of infected tissues (low pH, high-level H_2_O_2_, elevated enzymes, etc.) [99]. Any alteration in the bio milieu at the sick site is typically seen as a mediator for chemical or physical changes in the delivery system, which in turn stimulates the release of the drug or antibacterial payload. The release profile of medications is mostly determined by the physiological status of the illness site, making external manipulation of the target site problematic. External stimuli, on the other hand, such as light, sound, magnetism, and electrical activity, are examples of externally regulated systems (also known as open-loop systems). These systems can adjust the drug’s release profile transiently by varying the length and strength of the external stimulus, resulting in an exact supply of the medication at the desired dose [100]. 

Several studies demonstrated the stimuli-sensitive drug release. For pH, Luke E. Visscher et al. [101] created a 3D poly-caprolactone-tricalcium phosphate mesh for gentamicin sulphate distribution, while Mi et al. [102] described a novel zwitterionic hydrogel coupled with an antibacterial drug. On the other hand, Bhattacharyya et al. [103] used thin sol-gel films to control antibiotic delivery, demonstrating that the gel may limit MRSA and MSSA growth. However, the release profile of these carriers was storage and time-dependent. In another study, drug release with redox-sensitive systems was shown by Wang et al. [104] developed a system including polythioketal, selenium polymers, and aryalboronate to use ROS for targeted drug delivery. In another study by Thamphiwatana et al. [105] created liposomes carrying antibiotics such as doxycycline that were phospholipase A2 sensitive. Gold NPs were first stabilized using chitosan before being adsorbed onto liposomes. When these liposomes were exposed to the PLA2 enzyme of Helicobacter pylori, they released a large amount of antibiotics. Hence, targeted drug delivery at the proper time and place can considerably reduce adverse effects and resistance development, allowing for increased therapeutic efficacy and patient compliance.

### 3.4. Directed towards Biofilm Microenvironments

Most persistent human infectious diseases are caused by pathogenic biofilms, including those found in the mouth [106]. Due to the multi-component structure, biofilms provide a variety of purposes. Polysaccharides, proteins, nucleic acids, lipids, water, and ions (e.g., cations such as Ca^2+^ or Mg^2+^ and anions such as Cl or PO_4_^3−^) are the six categories of these components [107,108,109]. An approach used to lessen AMR, depending on the particular biofilm microenvironment, is that we can build stimuli-sensitive drug nanocarriers that take use of the biofilm milieu to boost delivery efficiency [110,111]. Veronica Folliero and her team conducted an experiment revealing that Indwelling Medical Devices (IMDs) not only serve as a source of persistent infections and a high risk of device-related infections (DRIs), but they also play a significant role in driving AMR. This finding underscores the importance of addressing biofilm colonization on IMDs to combat these challenges effectively [112].

In this regard, Alexander et al. [113] used a bioresponsive polymeric formulation generated from specially modified alginate NPs to administer ciprofloxacin (CIP) and the QSI, 3-amino-7-chloro-2-nonylquinazolin-4(3H)-one (ACNQ), to mature *Pseudomonas aeruginosa* biofilms. A pH-responsive linker was added between the polysaccharide backbone and ACNQ in the alginate NPs, and CIP was encapsulated via electrostatic interactions for a dual-action release of CIP and ACNQ in the low-pH state of a biofilm. When compared to CIP treatment alone, the collectively released CIP and ACNQ from pH-responsive NPs had a considerable killing effect on the biofilm.

According to certain studies, NPs affect bacteria in biofilms through the same processes as they affect planktonic bacteria. They include physical damage to bacteria caused by NPs sharp edges, [114] ion release, [115] the generation of reactive oxygen species (ROS) resulting in oxidative stress, [116] NP-mediated enzyme-like catalytic activity, [117] and DNA damage [118]. However, several experts have noted that the aforementioned methods do not adequately account for the intricate interactions between NPs and biofilms and that any new mechanism should take into account how the oral biofilm forms and is structured. 

### 3.5. Combined Physical Therapy

Magnetic NPs (MNs) have been coupled with other therapy modalities, such as PDT and PTT, to increase their effects is comes under combined physical therapy and employed as promising alternative treatments for a variety of diseases due to their distinct advantages, such as enhanced selectivity, low invasiveness, and adverse effects [119].

#### 3.5.1. Photothermal Therapy (PTT)

PTT, based on MNs under near-infrared (NIR) and laser radiation (ranging from 700 to 1100 nm), is a less invasive local therapeutic approach that not only directly kills diseased cells but also induces encapsulated medication release from the MNs, as it has the ability to deeply penetrate tissues [120]. Polymeric NPs, carbon-based nanomaterials [121], gold nanomaterials [122] metal sulfide nanomaterials (e.g., molybdenum disulfide (MoS_2_) and copper sulfide (CuS) [123,124] and organic compounds, such as organic dyes and polymers, have all demonstrated the ability to convert light to heat, which can be employed for PTT [16,125]. The efficacy of PTT-mediated bacterial mortality is affected by surface chemistry. For example, Justin and Chen [126]. Chitosan-reduced graphene oxide ((CS)-rGO) nanocomposites were created by combining graphene oxide (GO) suspension and chitosan solution in a biocompatible reduction process. The addition of rGO into CS may enhance the electrical conductivity and mechanical properties of the nanocomposite, making it suitable for biodegradable MN arrays as well as transdermal and intradermal drug delivery systems controlled by PTT or electroporation [126]. Zhao and co-workers modified CNTs-Fe_3_O_4_ using temperature-sensitive polymers; with NIR treatment, the resultant nano-agents changed from hydrophilic to hydrophobic and formed nanomaterial bacteria aggregates with localized heat effects on bacteria and increased NIR death. Furthermore, the bacteria can release nano-agents at low temperatures [127]. 

#### 3.5.2. Antibacterial Photodynamic Therapy (aPDT)

Due to the advent of AMR among dangerous bacteria, researchers have revitalized the antimicrobial effects of PDT and detonated aPDT in recent years. The mechanism is based on photosensitizer photochemical processes (PSs). When exposed to the light of the same wavelength as the PSs, hydroxyl radicals, superoxides, or singlet oxygen (^1^O_2_) ROS can be produced, causing damage to cell membranes and DNA molecules [16]. Porphyrins, chlorines, and phthalocyanines are the most often utilised PSs [128,129]. Compared to conventional antibiotic treatments and PTT, aPDT on localized superficial infections is more convenient and effective, with fewer side effects. According to Abrahamse and Hamblin’s report, several PS such as haematoporphyrin derivative, Photofrin, protoporphyrin IX, Verteporfin (benzoporphyrin derivative), Radachlorin (now Bremachlorin), fullerenes, Temoporfin, or Foscan (m-tetra-hydroxyphenyl-chlorin) has recently received clinical approval [130]. Several studies were employed to enhance the efficacy of NPs using PDT, for, e.g., recent research by Darabpour et al. [131], methylene blue can be encapsulated within chitosan NP to enhance the antibiofilm impact of aPDT with the PS (i.e., methylene blue) used [131]. Huang et al. effectively reported a similar technique, in which they employed ruthenium NP by adding cholinergic to its surface to improve its impact in both PDT and PTT, thereby eliminating MDR *Pseudomonas aeruginosa* [132]. Better activity enhancements by aPDT might be obtained not only by encapsulating or chemically integrating PS in/onto NP, but also by combining metal/metal oxide NP with PS, as described by Forzaneb et al. [133]. Because of the multi-target killing mechanism, the combination of aPDT and nanomaterial could be portrayed as a promising regime against resistant bacteria. 

## 4. Aspect of One Health with Nanotechnology and Economy

The idea of One Health applied to AMR has similarities with the use of nanotechnologies. The issue of AMR is a multi-faceted problem, which is primarily caused by the use of antimicrobials in human health, as well as the animal, environmental, and food industries, for various reasons and with varying degrees of necessity. AMR is evolving rapidly and spreading widely. Drug-resistant infections are mostly brought on by the improper use and overuse of antibiotics. Strategies such as using proper hand hygiene and cleaning the surroundings; triaging and isolating/cohorting patients with antibiotic-resistant illnesses; practicing antimicrobial stewardship; and undertaking surveillance may help to limit the impact of AMR [134]. The FAO, the OIE, and the WHO all concurred on the One Health concept [135]. Collaborations between the animal and public health sectors have been emphasized as a strategy to improve zoonotic danger management. Moreover, the rise of AMR will exacerbate the disparities between developing and developed nations, leading to a significant increase in inequity. This impact will predominantly affect individuals in low-income countries, as they are more susceptible to being pushed into extreme poverty as a consequence of AMR. The underprivileged populations in these countries will bear the brunt of the consequences, given their greater reliance on labor income, which will inevitably decline with a higher prevalence of infectious diseases caused by AMR [136]. AMR is a multi-faceted issue that needs to be addressed by several stakeholders. AMR’s strong global and national action plans as well as numerous stewardship initiatives should be strengthened with the development of communication strategies. For instance, a lack of interdepartmental coordination, ineffective implementation of regulations, inadequate support services, physician competition, and a lack of awareness, particularly at the policy making levels in developing countries, are both opportunities and challenges for experts in health communication [137,138]. Market impacts, such as trade and tourism rules and restrictions, as well as those resulting from consumer response and changes in consumer confidence in the food chain, are examples of such reactions [139].

Nanotechnology has emerged as a transformative force with significant direct impacts on the global economy. Through the creation of new industries and markets, improved manufacturing processes, and enhanced products, nanotechnology has opened doors for innovative solutions in various sectors [137,138]. The nanotechnology market’s global size reached approximately USD 85.39 billion in 2021, and projections indicate it will surpass USD 288.71 billion by 2030, exhibiting a Compound Annual Growth Rate (CAGR) of 14.5% from 2022 to 2030 [140]. Nanoparticles and nanoscale materials have been harnessed to create targeted drug delivery systems, improving drug efficacy while reducing side effects and toxicity. Furthermore, nanotechnology has facilitated the development of advanced imaging techniques, enabling early disease detection with higher precision [141]. Investing in nanotechnology research and development, along with education and skill development, will pave the way for sustainable economic development and innovation [142]. 

It is anticipated that by 2050, AMR will have killed 4,730,000 people in Asia alone [143], whereas AMR will cause 10 million annual deaths at the cost of USD 100 trillion, which will be equal to 7% GDP of world’s economy [13]. To comprehend the dynamics of resource use and advocate for change, economics is an essential discipline. One Health initiatives aim to improve disease risk mitigation by promoting a holistic approach, which is believed to be more efficient and effective than traditional sectorial approaches. In the context of Human and Animal Vaccination Delivery to Remote Nomadic Families in Chad, a notable case study demonstrates the significance of sustained vaccination programs as vital tools for both public health and the veterinary sectors. This approach effectively reduces operational costs associated with interventions requiring expensive transportation [144]. The advantage gained from increased human vaccination, resulting in savings for the human health care system. However, implementing such programs will require investments in human, institutional, and infrastructure development, and the resources are not always available instantly. Therefore, to assess the benefits of One Health, economic evaluations must compare the resource use and outcomes of the holistic approach with the existing practices.

According to the FAO, the predicted benefit of One Health to the global society in 2022 is at least USD 37 billion per year (FAO). The projected annual need for preventative spending is less than 10% of these benefits. Nevertheless, funding for One Health remains insufficient [145]. Investors in One Health must invest while considering two distinct factors: (1) The impact of disease: Disease costs in terms of cattle production losses, The cost of disease management, and human health consequences and costs. (2) Preventable losses—disease-related costs that can be avoided by establishing a disease control program. The disease effect provides an indication of the economic importance of a condition and whether further resources in terms of education and research are required. The choice on surveillance and action must consider if the expenses are less than the preventable losses [146]. Table 2 summarizes the integrating surveillance of AMR to a One Health level and the role of the integrated interventions of One Health surveillance.

## 5. Conclusions

Antibiotics are the foundations of modern medicine and have significantly contributed to the advancement of health care, but as AMR increased, it created a global threat to human life that has never been seen before. As a result, a nanotechnology-based drug delivery system for the production of future nanobiotics is viewed as a weapon in the twenty-first-century technological revolution, accomplished by building a governance structure to bring strategic and operational planning into alignment with one health approach. This gives optimism that the negative effects of antimicrobial resistance can be minimized and may not have irreversible consequences for society as a whole.

## Figures and Tables

**Figure 1 biomolecules-13-01182-f001:**
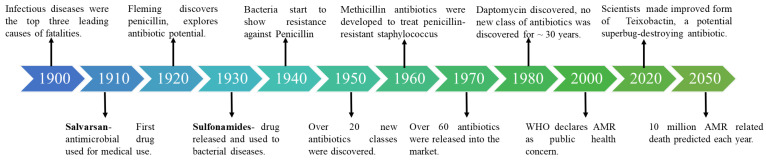
The antibiotic development timeline.

**Table 2 biomolecules-13-01182-t002:** Surveillance bodies formed by India and Taiwan to control AMR.

S.No.	Country	Organizations	Role
1.	India	National action Plan (NAP) https://iica.nic.in/sob_nap.aspx (accessed on 12 April 2023)	Implementation of a comprehensive and multi-sectorial NAP with a coordinated body, an operational strategy, and the creation of a monitoring system.
2.		National centre for diseases control (NCDC) https://ncdc.gov.in (accessed on 12 April 2023)	In collaboration with the State Governments, disease surveillance facilitates the prevention and control of communicable diseases.
3.		Antimicrobial Resistance Surveillance and Research Network (AMRSN) http://www.carss.cn (accessed on 12 April 2023)	Collect evidence and identify trends and patterns of drug-resistant diseases.
4.		Food safety and standard authority of India (FSSAI)	Collect and compile data on food consumption, contaminants in food and food product, identification of developing risks, and the implementation of a quick warning system.
5.	Taiwan	Taiwan centers for diseases control https://www.cdc.gov.tw/En (accessed on 12 April 2023)	Strengthening surveillance system, conduct hospital accreditation, run stewardship program.
6.		Infection control society of Taiwan https://nics.org.tw/ (accessed on 12 April 2023)	Formulating recommendations, engage in infection control activities in response to the government health policies.
7.		The infectious disease society of Taiwan (IDST) http://www.idsroc.org.tw/ (accessed on 12 April 2023)	Formation of rules and contributions in the prevention and control of antimicrobial resistance.
8.		Surveillance of Multicenter Antimicrobial Resistance in Taiwan (SMART)	Monitor the in vitro resistance of clinically important bacteria obtained from Taiwanese hospitals.
9.	U.S.A.	Food and Drug Administration (FDA) https://www.fda.gov/ (accessed on 12 April 2023)	Ensures public health through safeguarding the safety, effectiveness, and security of human and veterinary drugs, biological products, and medical devices.
10.		National Antimicrobial Resistance Monitoring System (NARMS) https://www.cdc.gov/narms/index.html (accessed on 12 April 2023)	Tracks changes in antimicrobial susceptibility of select foodborne enteric bacteria found in ill people, retail meats, and food animals.
11.		Pan American Health Organization (PAHO) https://www.paho.org/en/together-fight-antimicrobial-resistance (accessed on 12 April 2023)	Fostering strategic collaborations among Member States and other partners to promote health equity, fight against diseases, and enhance the quality of life and lifespan.
12.		Centers for Disease Control and Prevention https://www.cdc.gov/ (accessed on 12 April 2023)	As the nation’s health protection agency, CDC saves lives and protects people from health threats.
13.	Europe	European Antimicrobial Resistance Surveillance Network (EARS-Net) https://www.ecdc.europa.eu/en/ (accessed on 12 April 2023)	Play a crucial role in increasing awareness among political leaders, public health officials, the scientific community, and the general public.
14.		Healthcare-associated Infections Surveillance Network (HAI-Net) https://www.ecdc.europa.eu/en/ (accessed on 12 April 2023)	Provide baseline endemic infection rates by conducting surveillance.
15.		Organisation for Economic Co-operation and Development (OECD) https://www.oecd.org/els/health-systems/antimicrobial-resistance.htm (accessed on 12 April 2023)	Promote policies that cultivate prosperity, equality, opportunity, and well-being for everyone.
16.		Joint Programming Initiative on Antimicrobial Resistance (JPIAMR) https://www.jpiamr.eu/ (accessed on 12 April 2023)	Establish connections between research networks, research institutes/centers, and collaborate on implementing research focused on Antimicrobial Resistance (AMR) using a “One Health” approach.

## Data Availability

Not applicable.

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
