# Peer review of "Nanobiotics and the One Health Approach: Boosting the Fight against Antimicrobial Resistance at the Nanoscale"

_biomolecules, 2023, doi:10.3390/biom13081182_

Round 1

Reviewer 1 Report

In the current manuscript, Pandey et al. discuss nanotechnology for combating bacterial resistance. Although there are several review papers focusing on the same topic, the field changes rapidly and still deserves an update. However, several issues need to be addressed before the recommendation for publication.

1.      The authors are suggested to revise their abstract. Antimicrobial resistance (AMR) is caused by many factors, such as gene mutation, bacteria living in biofilms, and mammalian cells, as described by the authors in their main text. However, the authors stated that "the overuse and misuse of antibiotics have contributed significantly to the emergence and spread of antimicrobial resistance".

2.      The authors should discuss more on the reasons that cause antimicrobial resistance. Currently, only one sentence, "Bacteria, on the other hand,....posed by these antimicrobials." reach that question. Actually, bacterial biofilms and intracellular bacteria also contribute greatly to antimicrobial resistance due to drug delivery barriers. That is the main reason we need to develop nanocarriers to overcome those barriers and enhance the efficacy of conventional antibiotics. Some references may be helpful to the authors, for example, 10.1039/C7CS00807D and 10.1039/C7CS00748E.

3.      The references in the current manuscript need to be updated. Some stat-of-the-art references should be helpful to the authors' discussion, such as 10.1002/adma.202301623 and 10.1002/adfm.202214299 (drug combination).

4.      Section e on Page 8 was considerably short. The authors are suggested to extend this part.

5.      The definition of PTT, apart from near-infrared light, a laser with other wavelengths can also trigger PTT. The advantage of using near-infrared irradiation is that it can penetrate deep into tissues. 

Author Response

    We would like to thank you for your helpful and pertinent comments. Our point-by-point responses to your comments appear below.

Reviewer #1:In the current manuscript, Pandey et al. discuss nanotechnology for combating bacterial resistance. Although there are several review papers focusing on the same topic, the field changes rapidly and still deserves an update. However, several issues need to be addressed before the recommendation for publication.

  1. The authors are suggested to revise their abstract. Antimicrobial resistance (AMR) is caused by many factors, such as gene mutation, bacteria living in biofilms, and mammalian cells, as described by the authors in their main text. However, the authors stated that "the overuse and misuse of antibiotics have contributed significantly to the emergence and spread of antimicrobial resistance".

Ans- Thank you for pointing this out, the updated manuscript has been revised as suggested.

  1. The authors should discuss more on the reasons that cause antimicrobial resistance. Currently, only one sentence, "Bacteria, on the other hand,....posed by these antimicrobials." reach that question. Actually, bacterial biofilms and intracellular bacteria also contribute greatly to antimicrobial resistance due to drug delivery barriers. That is the main reason we need to develop nanocarriers to overcome those barriers and enhance the efficacy of conventional antibiotics. Some references may be helpful to the authors, for example, 10.1039/C7CS00807D and 10.1039/C7CS00748E.

Ans- We have expanded the discussion on the reasons causing antimicrobial resistance based on the valuable feedback provided. Thank you for your input.

  1. The references in the current manuscript need to be updated. Some stat-of-the-art references should be helpful to the authors' discussion, such as 10.1002/adma.202301623 and 10.1002/adfm.202214299 (drug combination).

Ans- We have taken your suggestion into consideration, and the references in the manuscript have been updated accordingly. We appreciate your input, and we believe that these updated references further strengthen the content of our manuscript.

  1. Section e on Page 8 was considerably short. The authors are suggested to extend this part.

Ans- Thank you for bringing this to our attention. PDT and PTT are the part of combination therapy and we have corrected the mistake in the updated manuscript. We apologize for any confusion this may have caused.

  1. The definition of PTT, apart from near-infrared light, a laser with other wavelengths can also trigger PTT. The advantage of using near-infrared irradiation is that it can penetrate deep into tissues.

Ans- Certainly! We have included it in the updated manuscript.

Reviewer 2 Report

The review article ‘Nanobiotics and the One Health Approach: Boosting the Fight against Antimicrobial Resistance at the Nanoscale’ provides a good overview of the economic aspects of One Health and its relationship with nanotechnology in addressing AMR. By incorporating specific examples, clarifying the relationship between nanotechnology and the economy, and addressing potential limitations, the passage can be further strengthened.

The review mentions the problem of antibiotic resistance and the socioeconomic impact of AMR but does not efficiently delve into the specific challenges and drawbacks associated with these issues. Exploring the complexities of AMR, such as the mechanisms of resistance, the role of different stakeholders, and the difficulties in implementing solutions, would provide a more comprehensive understanding of the topic.

It also does not acknowledge potential limitations or counterarguments against these approaches. Including a discussion of alternative strategies or potential criticisms would present a more balanced perspective.

General statements without supporting evidence: The introduction includes several statements, such as nanomaterials being "promising weapons" and the combination of nanotechnology and antibiotics being the most promising technique, without providing sufficient evidence or references to support these claims. Adding specific studies or research findings would strengthen the arguments made.

Overemphasis on economic development: While the introduction highlights the economic impact of AMR, it places a heavy focus on economic development as the main motivation to combat AMR. This perspective may overlook other important aspects, such as public health, ethical considerations, and environmental impacts. A more comprehensive analysis of the multidimensional effects of AMR would provide a more well-rounded discussion.

The passage would benefit from including specific examples or case studies to illustrate the economic benefits and costs associated with One Health approaches. This would enhance the reader's understanding and provide concrete evidence to support the claims made.

Clarify the relationship between nanotechnology and the economy: While the passage mentions nanotechnology in the context of One Health, it does not clearly explain the direct impact of nanotechnology on the economy. Elaborating on the economic implications of nanotechnology applications in AMR prevention and treatment would strengthen the argument.

It would be valuable to acknowledge and discuss more deeply about the potential limitations or challenges related to integrating One Health, nanotechnology, and the economy in combating AMR. This would provide a more balanced perspective and demonstrate a critical understanding of the topic.

The overall quality of English language in the passage is satisfactory, but there is room for improvement in certain areas. The passage generally demonstrates a good command of English, but there are a few areas where clarity and coherence could be enhanced. Some sentences may require rephrasing or restructuring to improve readability and flow. Additionally, there are instances where the use of punctuation, such as commas and hyphens, could be improved for better grammatical accuracy.

Author Response

The review article ‘Nanobiotics and the One Health Approach: Boosting the Fight against Antimicrobial Resistance at the Nanoscale’ provides a good overview of the economic aspects of One Health and its relationship with nanotechnology in addressing AMR. By incorporating specific examples, clarifying the relationship between nanotechnology and the economy, and addressing potential limitations, the passage can be further strengthened.

  1. The review mentions the problem of antibiotic resistance and the socioeconomic impact of AMR but does not efficiently delve into the specific challenges and drawbacks associated with these issues. Exploring the complexities of AMR, such as the mechanisms of resistance, the role of different stakeholders, and the difficulties in implementing solutions, would provide a more comprehensive understanding of the topic.

Ans- Thank you for bringing this to our attention. We have provided a more detailed discussion of the topic in the introduction section.

  1. It also does not acknowledge potential limitations or counterarguments against these approaches. Including a discussion of alternative strategies or potential criticisms would present a more balanced perspective.

Ans- Thank you for pointing this out, the updated manuscript has been revised as suggested.

  1. General statements without supporting evidence: The introduction includes several statements, such as nanomaterials being "promising weapons" and the combination of nanotechnology and antibiotics being the most promising technique, without providing sufficient evidence or references to support these claims. Adding specific studies or research findings would strengthen the arguments made.

Ans- Thank you for your suggestions. We have cited it with the supporting references.

  1. Overemphasis on economic development: While the introduction highlights the economic impact of AMR, it places a heavy focus on economic development as the main motivation to combat AMR. This perspective may overlook other important aspects, such as public health, ethical considerations, and environmental impacts. A more comprehensive analysis of the multidimensional effects of AMR would provide a more well-rounded discussion.

Ans- Thank you for your comment, with a thoroughly comprehensive discussion in the updated manuscript we have included a more comprehensive analysis of the multidimensional effects of AMR.

  1. The passage would benefit from including specific examples or case studies to illustrate the economic benefits and costs associated with One Health approaches. This would enhance the reader's understanding and provide concrete evidence to support the claims made.

Ans- We have taken your suggestion into consideration, and the references in the manuscript have been updated accordingly with supporting examples.

  1. Clarify the relationship between nanotechnology and the economy: While the passage mentions nanotechnology in the context of One Health, it does not clearly explain the direct impact of nanotechnology on the economy. Elaborating on the economic implications of nanotechnology applications in AMR prevention and treatment would strengthen the argument.

Ans- Thank you sincerely for your profound suggestion. We have incorporated the discussed points into the updated manuscript.

  1. It would be valuable to acknowledge and discuss more deeply about the potential limitations or challenges related to integrating One Health, nanotechnology, and the economy in combating AMR. This would provide a more balanced perspective and demonstrate a critical understanding of the topic.

Ans- Thank you for pointing this out, the updated manuscript has been revised as suggested.

Comments on the Quality of English Language

The overall quality of English language in the passage is satisfactory, but there is room for improvement in certain areas. The passage generally demonstrates a good command of English, but there are a few areas where clarity and coherence could be enhanced. Some sentences may require rephrasing or restructuring to improve readability and flow. Additionally, there are instances where the use of punctuation, such as commas and hyphens, could be improved for better grammatical accuracy.

Reviewer 3 Report

This review nicely synthesizes the current and potential future impact that nanotechnology can have on combatting AMR from both a technical and economic/societal standpoint. However, I think the impact of this review can be strengthened if the following comments are addressed:

1.     On page 2, regarding the UN sustainable development goals when the authors say, AMR is among the top priorities in achieving the 2030 UN Sustainable Development Goals is AMR actually listed formally as part of the Sustainable Development Goals? Clarify in the text if it is instead integrated into several of the goals without being explicitly mentioned.

2.     On page 2, the authors say, As a result, the One Health strategy to integrate AMR reduction will fail. What does this mean that it will fail and/or why? It seems contradictory to what the authors said previously.

3.     On page 2, the authors say, The most promising technique for dealing with AMR bacteria would be the combination of nanotechnology and antibiotics. Can the authors explain in the text why the combination is better instead of nanotechnology alone where certain nanomaterials can deliver their own mechanisms of antimicrobial activity? Further, with the combination approach, is the goal to rescue the antibiotic and allow it to once again be effective and/or allow organisms to regain sensitivity to traditional antibiotics?

a.     In a short section, the authors should discuss mechanisms of antimicrobial activity of the nanomaterials alone, for example, silver ion release is largely responsible for antimicrobial activity of silver nanoparticles. The authors could mention that these nanomaterials can be used as antimicrobial surface coatings rather than administered to humans to ingest?

4.     On page 2, the authors say, Many antibiotics were created in the twentieth century to treat bacterial infections. As a result, microorganisms adapted to growing amounts of antibiotics in

the environment in response to AMR. If the authors agree that this suggestion would increase the impact of their review, consider including a figure that outlines the historical timeline of when different antibiotics were developed and when first instances of bacterial resistance to those antibiotics started rising to support this statement.

5.     The authors should consider discussing the idea of cross/co-resistance against different antibiotic drug classes and heavy metals (via general resistance mechanisms) they could include a short section in the text dedicated to this topic.

6.     The authors should explain slightly more thoroughly why antibiotic use in animal agriculture is a problem and is part of the antibiotic misuse by humans discussion transmission pathways from agriculture/environment to humans. Also, detail how doctors mis-prescribing antibiotics for viral infections and how patients not taking the full antibiotic dosing regimen can exacerbate AMR. The opposite is true too when patients take the full antibiotic dosing regimen (when not necessary), sometimes this promotes further resistance to develop in an individuals gut.

a.     Further, how do organisms get exposed/adapt to growing amounts of antibiotics in the environment? How do antibiotics end up in the environment?

7.     The introduction and antibiotics and AMR sections have some redundancy/overlapping content about AMR and infectious disease. Please consider removing some of the redundancies to make these sections more concise.

8.     On page 3, the following sentences are redundant and should be condensed: One commonly acknowledged connection between nanomaterials and their ability to fight bacteria is that they show promise as a supplement to antibiotics. This approach is becoming increasingly popular as it has the potential to address the limitations of antibiotics. Furthermore, as an excellent transporter, nanomaterials can supplement and assist existing antibiotics.

9.     In figure 1, antibiotic resistance is labeled but the text refers to it as antimicrobial resistance (AMR). Choose one and keep terminology consistent throughout the manuscript and figures.

10.   Figure 1 is a little confusing - how are all parts of this figure showing different delivery mechanisms? This figure looks like it illustrates different interaction mechanisms (in part ii) but not necessarily conveying or illustrating different delivery mechanisms in all parts of the figure (maybe in part iii and iv). Either re-word the text/explanation in the manuscript to describe these as mechanisms of interaction too or update the figure to show a greater variety of delivery mechanisms. Part i of this figure also doesnt seem to relate to delivery or interaction mechanisms.

11.   The authors should discuss biofilms in the context of AMR infections (e.g., related to catheters and other medical devices). How does oral biofilm in the mouth relate to AMR? Is this a big source of AMR?

12.   On page 6, define MNPs the first time they appear in the text.

13.   On page 7, the authors state, On the other side, we can remold the biofilm microenvironment to improve antibiotic bactericidal activity. Can the authors provide examples or an explanation as to how the biofilm microenvironment can be remolded? What specific aspects or stages of biofilm formation/maturation can be remolded and how? What stages of biofilm formation are being impacted or have greatest potential for NP targeted use? The authors mention penetration of NPs but this is not exactly explaining how the biofilm is being remolded.

a.     Further, how do you ensure penetration of NPs into a biofilm? By using a certain/small enough particle size for example?

14.   The authors should include citations for the following statements:

a.     Page 7/8: However, several experts have noted that the aforementioned methods do not adequately account for the intricate interactions between NPs and biofilms and that any new mechanism should take into account how the oral biofilm forms and is structured.

b.     Page 10: Due to the advent of AMR among dangerous bacteria, researchers have revitalized the antimicrobial effects of PDT and detonated aPDT in recent years.

c.     According to the FAO, the predicted benefit of One Health to the global society in 2022 is at least USD 37 billion per year (FAO).

d.     The projected annual need for preventative spending is less than 10% of these benefits.

15.   The authors should move section e: combined therapy to after PDT and PTT sections for logical flow.

16.   On page 8, the authors say, Furthermore, the bacteria can release nano-agents at low temperatures for reusable disinfection. Can the authors elaborate on re-usability as a benefit of nano? What do the authors mean by reusable disinfection here? Do the authors mean nano surface coatings for disinfection dont need continually re-applied like traditional cleaning disinfectants? Or do they mean obtaining nano carriers from the initial application for re-use again in the same/different application? If so, how can you obtain nano carriers from the application to use again?

a.     Further, are nano carriers toxic to release inside and accumulate in human tissue? For example, should silver nanoparticles be ingested by humans or reserved as surface coatings?

17.   For table 1,

a.     Table 1 does not seem to include PS / aPDT even though the authors say it is summarized in Table 1 on page 10. Should these be included or else remove the sentence saying it is summarized in Table 1.

b.     Table 1 should be expanded a bit more to contain more information and more studies. Multiple references could easily be included and should be included for each category to be comprehensive and impactful as a review.

                                               i.     Nanoparticle size, shape, surface chemistry should be included as factors for all categories of nanomaterials, especially silver nanoparticles. Include more studies to reference for this. Example but not limited to:

                                              ii.     Stabryla, Lisa M., et al. "Emerging investigator series: it's not all about the ion: support for particle-specific contributions to silver nanoparticle antimicrobial activity." Environmental Science: Nano 5.9 (2018): 2047-2068.

                                             iii.     Johnston, Kathryn A., et al. "Impacts of broth chemistry on silver ion release, surface chemistry composition, and bacterial cytotoxicity of silver nanoparticles." Environmental Science: Nano 5.2 (2018): 304-312.

                                             iv.     Graf, Christina, et al. "Shape-dependent dissolution and cellular uptake of silver nanoparticles." Langmuir 34.4 (2018): 1506-1519.

c.     The carbon-based category should be expanded to capture reduced graphene moieties/graphene oxide/graphene sheets, CNTs that could be multi or single walled, etc. Surface chemistry is a factor affecting antimicrobial activity for carbon materials, as the authors mention earlier, and this should be stated accordingly in the factor column for carbon materials.

d.     Why are only certain bacteria targeted with each nanomaterial category? Can the authors explain this in the text? Certainly any material can be used to target any bacteria why are the authors limiting to specific bacteria per category? Is there a reason why these materials work better in targeting the certain bacteria listed? Or is it that only the studies listed have demonstrated effectiveness of these materials against those bacteria? Please explain.

e.     The authors should give an example of the combined composite matrixes for each category where does CNTs-Fe3O4 fall?

18.   On page 10, the authors say, The phenomenon is evolving rapidly and spreading widely what phenomenon are the authors referring to here? Be specific.

19.   On page 10, in section 4, I like that the authors mention varying degrees of necessity regarding antimicrobial use across the different sectors, but they should expand upon this idea and explicitly caution against the ubiquitous use of antimicrobial agents in applications that do not necessitate antimicrobial function and specifically recommend/propose they are reserved for health care applications.

a.     The authors should further discuss the necessity of using nanotechnology (and/or combination of nano/antibiotics) and which applications should be using/implementing/integrating nanotechnology/combined therapies, e.g., reserve for only health care applications such that they are effective (and have minimal resistance) when we desperately need them to work for patients with infections?

1.     In a brief section, the authors should also discuss the problematic idea of bacterial resistance developing towards even promising nanotechnologies, as shown/demonstrated in these papers specific to silver nanoparticles:

a.     Stabryla, L.M., Johnston, K.A., Diemler, N.A. et al. Role of bacterial motility in differential resistance mechanisms of silver nanoparticles and silver ions. Nat. Nanotechnol. 16, 9961003 (2021). https://doi.org/10.1038/s41565-021-00929-w

b.     Panáček, A., Kvítek, L., Smékalová, M. et al. Bacterial resistance to silver nanoparticles and how to overcome it. Nature Nanotech 13, 6571 (2018). https://doi.org/10.1038/s41565-017-0013-y

c.     Graves Jr, Joseph L., et al. "Rapid evolution of silver nanoparticle resistance in Escherichia coli." Frontiers in genetics 6 (2015): 42.

d.     McNeilly, Oliver, et al. "Emerging concern for silver nanoparticle resistance in Acinetobacter baumannii and other bacteria." Frontiers in Microbiology 12 (2021): 652863.

e.     Ewunkem, Akamu J., et al. "Experimental evolution of magnetite nanoparticle resistance in Escherichia coli." Nanomaterials 11.3 (2021): 790.

20.   The economic considerations are an interesting aspect to discuss in context of AMR. However, the focus presented here in the text is on zoonotic disease the authors should bring back this discussion to AMR and connect it to AMR to remain focused on the central point of AMR, instead of jumping around to zoonotic diseases. Is animal to human transmission a big pathway for AMR in humans? Maybe discuss this in the transmission pathways section mentioned above.

21.   On page 11, I like that the authors discuss infrastructure and education as essential aspects of AMR. Can the authors elaborate on/propose what infrastructure development/resources are needed to implement such programs? For example, education of farmers/stakeholders in agriculture industry on antibiotic use, education of doctors/patients, etc?

22.   In Table 2, action plans in the US and Europe/UN-centric should be included as well.

a.     Please mention if any of these groups are doing zoonotic surveillance since it was discussed earlier in the text?

minor grammatical errors/sentence construction - some sentences are fragments and not really sentences. make sure verb choice/tense matches noun.

Author Response

Thank you for taking the time to review our manuscript and for providing us with your helpful and pertinent comments. Your expertise and insights have been invaluable in improving the quality of our work.

This review nicely synthesizes the current and potential future impact that nanotechnology can have on combatting AMR from both a technical and economic/societal standpoint. However, I think the impact of this review can be strengthened if the following comments are addressed:

  1. On page 2, regarding the UN sustainable development goals when the authors say, “AMR is among the top priorities in achieving the 2030 UN Sustainable Development Goals” – is AMR actually listed formally as part of the Sustainable Development Goals? Clarify in the text if it is instead integrated into several of the goals without being explicitly mentioned.

Ans- Thank you for your suggestion. We have clarified the statement in the updated manuscript.

  1. On page 2, the authors say, “As a result, the One Health strategy to integrate AMR reduction will fail.” What does this mean that it will fail and/or why? It seems contradictory to what the authors said previously.

Ans- Thank you for pointing this out, we have removed the contradictory statement.

  1. On page 2, the authors say, “The most promising technique for dealing with AMR bacteria would be the combination of nanotechnology and antibiotics.” Can the authors explain in the text why the combination is better instead of nanotechnology alone where certain nanomaterials can deliver their own mechanisms of antimicrobial activity? Further, with the combination approach, is the goal to rescue the antibiotic and allow it to once again be effective and/or allow organisms to regain sensitivity to traditional antibiotics?
    1. In a short section, the authors should discuss mechanisms of antimicrobial activity of the nanomaterials alone, for example, silver ion release is largely responsible for antimicrobial activity of silver nanoparticles. The authors could mention that these nanomaterials can be used as antimicrobial surface coatings rather than administered to humans to ingest?

Ans- Thank you sincerely for your profound suggestion. We have taken it into serious consideration and have now incorporated the discussed points into the updated manuscript.

  1. On page 2, the authors say, “Many antibiotics were created in the twentieth century to treat bacterial infections. As a result, microorganisms adapted to growing amounts of antibiotics in the environment in response to AMR. If the authors agree that this suggestion would increase the impact of their review, consider including a figure that outlines the historical timeline of when different antibiotics were developed and when first instances of bacterial resistance to those antibiotics started rising to support this statement.

Ans- We have included a figure outlining the historical timeline of when different antibiotics were developed and when first instances of bacterial resistance to those antibiotics started rising to support this statement.

  1. The authors should consider discussing the idea of cross/co-resistance against different antibiotic drug classes and heavy metals (via general resistance mechanisms) – they could include a short section in the text dedicated to this topic.

Ans- We greatly appreciate the idea you shared, and we have taken it into careful consideration. Consequently, we have included the suggested topic in the updated manuscript.

  1. The authors should explain slightly more thoroughly why antibiotic use in animal agriculture is a problem and is part of the antibiotic misuse by humans – discussion transmission pathways from agriculture/environment to humans. Also, detail how doctors misprescribing antibiotics for viral infections and how patients not taking the full antibiotic dosing regimen can exacerbate AMR. The opposite is true too – when patients take the full antibiotic dosing regimen (when not necessary), sometimes this promotes further resistance to develop in an individual’s gut.
    1. Further, how do organisms get exposed/adapt to growing amounts of antibiotics in the environment? How do antibiotics end up in the environment?

Ans- Thank you for your valuable idea. We sincerely appreciate it, and we have duly incorporated the suggestions into the updated manuscript.

  1. The “introduction” and “antibiotics and AMR” sections have some redundancy/overlapping content about AMR and infectious disease. Please consider removing some of the redundancies to make these sections more concise.

Ans- Thank you for pointing this out, we have removed the redundancy/overlapping content from both the sections.

  1. On page 3, the following sentences are redundant and should be condensed: “One commonly acknowledged connection between nanomaterials and their ability to fight bacteria is that they show promise as a supplement to antibiotics. This approach is becoming increasingly popular as it has the potential to address the limitations of antibiotics. Furthermore, as an excellent transporter, nanomaterials can supplement and assist existing antibiotics”.

Ans- Thank you for pointing this out, we have removed the redundancy/overlapping statement from the sentences.

  1. In figure 1, antibiotic resistance is labeled but the text refers to it as antimicrobial resistance (AMR). Choose one and keep terminology consistent throughout the manuscript and figures.

Ans- Thank you for pointing this out, we have kept the terminology consistent throughout the updated manuscript and figures

  1. Figure 1 is a little confusing - how are all parts of this figure showing different delivery mechanisms? This figure looks like it illustrates different interaction mechanisms (in part ii) but not necessarily conveying or illustrating different delivery mechanisms in all parts of the figure (maybe in part iii and iv). Either re-word the text/explanation in the manuscript to describe these as mechanisms of interaction too or update the figure to show a greater variety of delivery mechanisms. Part i of this figure also doesn’t seem to relate to delivery or interaction mechanisms.

Ans- Thank you for the suggestion. We have re-word the text/explanation in the revised manuscript.

  1. The authors should discuss biofilms in the context of AMR infections (e.g., related to catheters and other medical devices). How does oral biofilm in the mouth relate to AMR? Is this a big source of AMR?

Ans- In the updated manuscript, we have thoroughly explored the topic of biofilms in the context of AMR infections.

  1. On page 6, define MNPs the first time they appear in the text.

Ans- Certainly! We have defined the MNPs in the updated manuscript.

  1. On page 7, the authors state, “On the other side, we can remold the biofilm microenvironment to improve antibiotic bactericidal activity.” Can the authors provide examples or an explanation as to how the biofilm microenvironment can be remolded? What specific aspects or stages of biofilm formation/maturation can be remolded and how? What stages of biofilm formation are being impacted or have greatest potential for NP targeted use? The authors mention penetration of NPs but this is not exactly explaining how the biofilm is being remolded.
    1. Further, how do you ensure penetration of NPs into a biofilm? By using a certain/small enough particle size for example?

Ans- Thank you for pointing this out, we have removed this line from the updated manuscript.

  1. The authors should include citations for the following statements:
    1. Page 7/8: “However, several experts have noted that the aforementioned methods do not adequately account for the intricate interactions between NPs and biofilms and that any new mechanism should take into account how the oral biofilm forms and is structured.”
    2. Page 10: “Due to the advent of AMR among dangerous bacteria, researchers have revitalized the antimicrobial effects of PDT and detonated aPDT in recent years.
    3. According to the FAO, the predicted benefit of One Health to the global society in 2022 is at least USD 37 billion per year (FAO).
    4. The projected annual need for preventative spending is less than 10% of these benefits.

Ans-We have taken your suggestion into consideration, and the references in the manuscript have been updated accordingly. We appreciate your input, and we believe that these updated references further strengthen the content of our manuscript.

  1. The authors should move section “e: combined therapy” to after PDT and PTT sections for logical flow.

Ans- Thank you for bringing this to our attention. PDT and PTT are the part of combination therapy and we have corrected the mistake in the updated manuscript. We apologize for any confusion this may have caused.

  1. On page 8, the authors say, “Furthermore, the bacteria can release nano-agents at low temperatures for reusable disinfection.” Can the authors elaborate on re-usability as a benefit of nano? What do the authors mean by reusable disinfection here? Do the authors mean nano surface coatings for disinfection don’t need continually re-applied like traditional cleaning disinfectants? Or do they mean obtaining nano carriers from the initial application for re-use again in the same/different application? If so, how can you obtain nano carriers from the application to use again?
    1. Further, are nano carriers toxic to release inside and accumulate in human tissue? For example, should silver nanoparticles be ingested by humans or reserved as surface coatings?

Ans- Thank you for pointing this out, as it was a typological error we have removed the words ‘for reusable disinfection’ in the updated manuscript.

  1. For table 1,
    1. Table 1 does not seem to include PS / aPDT even though the authors say it is summarized in Table 1 on page 10. Should these be included or else remove the sentence saying it is summarized in Table 1.
    2. Table 1 should be expanded a bit more to contain more information and more studies. Multiple references could easily be included and should be included for each category to be comprehensive and impactful as a review.
      1. Nanoparticle size, shape, surface chemistry should be included as factors for all categories of nanomaterials, especially silver nanoparticles. Include more studies to reference for this. Example but not limited to:
      2. Stabryla, Lisa M., et al. "Emerging investigator series: it's not all about the ion: support for particle-specific contributions to silver nanoparticle antimicrobial activity." Environmental Science: Nano 5.9 (2018): 2047-2068.
  • Johnston, Kathryn A., et al. "Impacts of broth chemistry on silver ion release, surface chemistry composition, and bacterial cytotoxicity of silver nanoparticles." Environmental Science: Nano 5.2 (2018): 304-312.
  1. Graf, Christina, et al. "Shape-dependent dissolution and cellular uptake of silver nanoparticles." Langmuir 34.4 (2018): 1506-1519.

Ans- We have expanded the table 1 with multiple references for each category to increase the impact.

  1. The carbon-based category should be expanded to capture reduced graphene moieties/graphene oxide/graphene sheets, CNTs that could be multi or single walled, etc. Surface chemistry is a factor affecting antimicrobial activity for carbon materials, as the authors mention earlier, and this should be stated accordingly in the “factor” column for carbon materials.

Ans- Thank you for this suggestion. It would have been interesting to explore this aspect. In this case, our focus is not centered on a specific nanoparticle; rather, we aim to provide comprehensive information on the potential of nanoparticles in general. Thus, delving into a single nanoparticle might slightly exceed the current scope of our research.

  1. Why are only certain bacteria targeted with each nanomaterial category? Can the authors explain this in the text? Certainly any material can be used to target any bacteria – why are the authors limiting to specific bacteria per category? Is there a reason why these materials work better in targeting the certain bacteria listed? Or is it that only the studies listed have demonstrated effectiveness of these materials against those bacteria? Please explain.

Ans- Yes, these nanomaterial’s can also work on other bacteria apart from the list, in the studies listed have demonstrated effectiveness of these materials against those bacteria. We have incorporated the bacteria in the specific table.

  1. The authors should give an example of the combined composite matrixes for each category – where does CNTs-Fe3O4 fall?

Ans- Thank you for the suggestion. While it may have been interesting to explore this aspect further, adding an example for every composite matrix in the table could make it difficult to follow and might not provide a clear perspective on each category. Instead, we can focus on providing concise and relevant information for each category, ensuring that the table remains easy to understand and informative.

  1. On page 10, the authors say, “The phenomenon is evolving rapidly and spreading widely” – what phenomenon are the authors referring to here? Be specific.

Ans- Thank you for bringing this to our attention. Here the phenomenon is referred to Antimicrobial resistance. We have updated in the manuscript as well.

  1. On page 10, in section 4, I like that the authors mention “varying degrees of necessity” regarding antimicrobial use across the different sectors, but they should expand upon this idea and explicitly caution against the ubiquitous use of antimicrobial agents in applications that do not necessitate antimicrobial function and specifically recommend/propose they are reserved for health care applications.
    1. The authors should further discuss the necessity of using nanotechnology (and/or combination of nano/antibiotics) and which applications should be using/implementing/integrating nanotechnology/combined therapies, e.g., reserve for only health care applications such that they are effective (and have minimal resistance) when we desperately need them to work for patients with infections?

Ans- We have considered your suggestion and expanded upon this idea and explicitly caution against the ubiquitous use of antimicrobial agents in applications that do not necessitate antimicrobial function in the updated manuscript.

  1. In a brief section, the authors should also discuss the problematic idea of bacterial resistance developing towards even promising nanotechnologies, as shown/demonstrated in these papers specific to silver nanoparticles:
    1. Stabryla, L.M., Johnston, K.A., Diemler, N.A. et al. Role of bacterial motility in differential resistance mechanisms of silver nanoparticles and silver ions. Nat. Nanotechnol. 16, 996–1003 (2021). https://doi.org/10.1038/s41565-021-00929-w
    2. Panáček, A., Kvítek, L., Smékalová, M. et al. Bacterial resistance to silver nanoparticles and how to overcome it. Nature Nanotech 13, 65–71 (2018). https://doi.org/10.1038/s41565-017-0013-y
    3. Graves Jr, Joseph L., et al. "Rapid evolution of silver nanoparticle resistance in Escherichia coli." Frontiers in genetics 6 (2015): 42.
    4. McNeilly, Oliver, et al. "Emerging concern for silver nanoparticle resistance in Acinetobacter baumannii and other bacteria." Frontiers in Microbiology 12 (2021): 652863.
    5. Ewunkem, Akamu J., et al. "Experimental evolution of magnetite nanoparticle resistance in Escherichia coli." Nanomaterials 11.3 (2021): 790.

Ans- Thank you for this suggestion. It would have been interesting to explore this aspect. However, in the case of our study, it seems slightly out of scope. As we are working on the manuscript to demonstrate the potential of nanobiotics in combating AMR, we recognize that placing too much emphasis on silver nanoparticles may lead to unnecessary lengthening, as discussed in the provided table. Hence, we aim to strike a balance in our discussion, highlighting the role of nanobiotics, including silver nanoparticles, while ensuring the manuscript remains concise and focused.

  1. The economic considerations are an interesting aspect to discuss in context of AMR. However, the focus presented here in the text is on zoonotic disease – the authors should bring back this discussion to AMR and connect it to AMR to remain focused on the central point of AMR, instead of jumping around to zoonotic diseases. Is animal to human transmission a big pathway for AMR in humans? Maybe discuss this in the transmission pathways section mentioned above.

Ans- Thank you for your valuable input. We have taken your suggestions into account and have duly incorporated them into the updated manuscript. In doing so, we ensured a clearer focus on AMR while addressing the economic considerations. The updated manuscript now provides a more comprehensive and focused discussion on the central point of AMR.

  1. On page 11, I like that the authors discuss infrastructure and education as essential aspects of AMR. Can the authors elaborate on/propose what infrastructure development/resources are needed to implement such programs? For example, education of farmers/stakeholders in agriculture industry on antibiotic use, education of doctors/patients, etc?

Ans- Thank you for appreciating our discussion on the importance of infrastructure and education in tackling AMR on page 11 of the manuscript. We agree that it is crucial to provide more details on the required infrastructure and resources to implement effective AMR programs.

In the introduction section, we have indeed elaborated on the significance of infrastructure development and education in various sectors related to AMR. These programs would focus on responsible antibiotic use, prudent prescribing practices, and raising awareness about the consequences of AMR.

  1. In Table 2, action plans in the US and Europe/UN-centric should be included as well.
    1. Please mention if any of these groups are doing zoonotic surveillance since it was discussed earlier in the text?

Ans- Thank you for your suggestion. We have added relevant information in the table, on whether any of these groups are actively involved in zoonotic surveillance.

Round 2

Reviewer 2 Report

Based on the reviewer's feedback, it appears that the responses provided by the authors are somewhat generic and do not fully address the specific concerns raised. Especially, The authors mentioned that they have cited supporting references for the general statements in the introduction. To satisfy the reviewer's concern, they should list the references they have included and briefly explain how these references provide evidence for the claims made in the article.

The quality of English language in the responses is improved, but with a few additional adjustments, the authors can further elevate the clarity and effectiveness of their communication.

Author Response

The review article ‘Nanobiotics and the One Health Approach: Boosting the Fight against Antimicrobial Resistance at the Nanoscale’ provides a good overview of the economic aspects of One Health and its relationship with nanotechnology in addressing AMR. By incorporating specific examples, clarifying the relationship between nanotechnology and the economy, and addressing potential limitations, the passage can be further strengthened.

  1. The review mentions the problem of antibiotic resistance and the socioeconomic impact of AMR but does not efficiently delve into the specific challenges and drawbacks associated with these issues. Exploring the complexities of AMR, such as the mechanisms of resistance, the role of different stakeholders, and the difficulties in implementing solutions, would provide a more comprehensive understanding of the topic.

Ans- Thank you for bringing this to our attention. We have provided a more detailed discussion of the topic in the introduction section.

  • The mechanisms of resistance is expended in the introduction section ‘Apart from that, bacterial isolates…… environmental settings into clinical environments’ with the proper reference including:
  1. https://doi.org/10.1007/S11356-017-8364-3
  2. https://doi.org/10.1038/ismej.2016.155
  3. https://doi.org/10.1016/j.envpol.2017.09.084
  4. https://doi.org/10.1007/S11356-017-8364-3
  • Furthermore, the role of different stakeholders were included in the same section ‘Antimicrobial use involves various stakeholders, …… play a significant role’ cited by https://doi.org/10.1007/S11356-017-8364-3
  • Additionally, the difficulties in implementing solutions has been discussed in the ‘Aspect of One health with nanotechnology and economy’ section, ‘AMR is a multifaceted issue that needs to be addressed by several stakeholders. …… for experts in health communication’.

https://doi.org/10.1016%2Fj.onehlt.2020.100171, https://doi.org/10.3389/fpubh.2020.493904.

  1. It also does not acknowledge potential limitations or counterarguments against these approaches. Including a discussion of alternative strategies or potential criticisms would present a more balanced perspective.

Ans- Thank you for pointing this out, the updated manuscript has been revised as suggested.

‘Implementing ASP comes with a set of challenges…... due to unavailability of certain antibiotics’ (https://doi.org/10.1017/ASH.2022.359)

This sentence will provide the different potential limitations ‘Many of the alternative strategies …….. unconventional strategies’

(https://doi.org/10.3390/ANTIBIOTICS11020200)

This added line further may help to introduce alternative strategies or potential criticisms of antimicrobial resistance has been discussed in section 4. ‘Aspect of One health with nanotechnology and economy’.

  1. General statements without supporting evidence: The introduction includes several statements, such as nanomaterials being "promising weapons" and the combination of nanotechnology and antibiotics being the most promising technique, without providing sufficient evidence or references to support these claims. Adding specific studies or research findings would strengthen the arguments made.

Ans- Thank you for your suggestions. We have cited it with the supporting references. For ‘nanomaterials being “promising weapons”’ (http://dx.doi.org/10.3390/nano11123346) It helps to understand how nanotechnology will become more widely involved in the diagnosis and treatment of diseases in the future, potentially helping to overcome bottlenecks under existing medical methods 

To provide the evidence for ‘the combination of nanotechnology and antibiotics being the most promising technique’ we have provide the reference as: (https://doi.org/10.3390%2Fantibiotics10121473) It help to understand the antimicrobial effects of nanoparticles and contrasts nanoparticles’ with antibiotics’ role in the fight against pathogenic microorganisms.

  1. Overemphasis on economic development: While the introduction highlights the economic impact of AMR, it places a heavy focus on economic development as the main motivation to combat AMR. This perspective may overlook other important aspects, such as public health, ethical considerations, and environmental impacts. A more comprehensive analysis of the multidimensional effects of AMR would provide a more well-rounded discussion.

Ans- Thank you for your comment, with a thoroughly comprehensive discussion in the updated manuscript we have included a more comprehensive analysis of the multidimensional effects of AMR.

To provide a well round discussion we have elaborated with the following paragraph:  ‘Antibiotic resistance poses significant public health concerns … and selected due to the use of antibiotics in humans’.

The following references support the provided information:

  • https://doi.org/10.3390/MOLECULES23040795
  • https://doi.org/10.1093/FEMSEC/FIW023
  • https://doi.org/10.3389/FMICB.2014.00284/BIBTEX
  • https://doi.org/10.1073/PNAS.092142499.

  1. The passage would benefit from including specific examples or case studies to illustrate the economic benefits and costs associated with One Health approaches. This would enhance the reader's understanding and provide concrete evidence to support the claims made.

Ans- We have taken your suggestion into consideration, and the references in the manuscript have been updated accordingly with supporting examples.

A case study is included in the manuscript providing insight view of Human and Animal Vaccination Delivery program to Remote Nomadic Families in Chad under the One Health approach, offers valuable insights into the program's effectiveness and economic impact. (https://doi.org/10.3201%2Feid1303.060391)

  1. Clarify the relationship between nanotechnology and the economy: While the passage mentions nanotechnology in the context of One Health, it does not clearly explain the direct impact of nanotechnology on the economy. Elaborating on the economic implications of nanotechnology applications in AMR prevention and treatment would strengthen the argument.

Ans- Thank you sincerely for your profound suggestion. We have incorporated the discussed points into the updated manuscript.

To understand more clearly the economic implications of nanotechnology applications in AMR prevention and treatment we have discussed in the following passage: ‘Nanotechnology has emerged as a ….. will pave the way for sustainable economic development and innovation.

These references corroborate the given information:

  • https://pubs.acs.org/doi/10.1021/acsnano.1c10919
  • https://doi.org/10.3390/MOLECULES28020661,
  • https://www.bccresearch.com/market-research/nanotechnology/global-nanotechnology-market.html
  • https://pubs.acs.org/doi/10.1021/acsnano.1c10919.

  1. It would be valuable to acknowledge and discuss more deeply about the potential limitations or challenges related to integrating One Health, nanotechnology, and the economy in combating AMR. This would provide a more balanced perspective and demonstrate a critical understanding of the topic.

Ans- Thank you for pointing this out, the updated manuscript has been revised as suggested.

The various challenges and limitations have been discussed in the introduction:

Antimicrobial resistance is a multifaceted issue … challenges for experts in health communication.

These references confirm the provided information:

  • https://doi.org/10.1016%2Fj.onehlt.2020.100171
  • https://doi.org/10.3389/fpubh.2020.493904.

Furthermore, to delve deeper into the topic we have further discussed in the section 4 ‘Aspect of One health with nanotechnology and economy

Moreover, the rise AMR will exacerbate the disparities …… which will inevitably decline with a higher prevalence of infectious diseases caused by AMR.

The following references support the provided information:

  • https://doi.org/10.2147%2FIDR.S234610
  • https://doi.org/10.2147/IDR.S234610

Comments on the Quality of English Language

The overall quality of English language in the passage is satisfactory, but there is room for improvement in certain areas. The passage generally demonstrates a good command of English, but there are a few areas where clarity and coherence could be enhanced. Some sentences may require rephrasing or restructuring to improve readability and flow. Additionally, there are instances where the use of punctuation, such as commas and hyphens, could be improved for better grammatical accuracy.

Ans- In the updated manuscript we have improved the sentences, improved punctuation, and enhanced readability make the text more engaging.